# Dispersal Limitation Controlling the Assembly of the Fungal Community in Karst Caves

**DOI:** 10.3390/jof9101013

**Published:** 2023-10-13

**Authors:** Zhi-Feng Zhang, Jian Mao, Lei Cai

**Affiliations:** 1Southern Marine Science and Engineering Guangdong Laboratory (Guangzhou), Guangzhou 511458, China; zhangzf@gmlab.ac.cn; 2State Key Laboratory of Mycology, Institute of Microbiology, Chinese Academy of Sciences, Beijing 100101, China; 3College of Life Sciences, University of Chinese Academy of Sciences, Beijing 100049, China

**Keywords:** fungal community, Karst cave, deterministic processes, environmental selection, stochastic processes

## Abstract

As a unique ecosystem, Karst caves harbor an impressive diversity of specific fungi. However, the factors and mechanisms that shape fungal biodiversity in caves remain elusive. In this study, we explored the assembly patterns of fungal communities based on our previous research in eight representative Karst caves in Southwest China. Our results indicated that dispersal limitation plays a crucial role in shaping the overall fungal community as well as specific communities in rock, sediment, and water samples. However, “Undominated” processes contributed more than dispersal limitation in air samples. Interestingly, the dominant assembly processes varied between caves. Consistently, environmental selection had a minor impact on the assembly of fungal communities. Among the examined spatial and environmental variables, latitude, longitude, altitude, and temperature were found to significantly influence fungal communities irrespective of substrate type. These findings provide valuable insights into the ecological factors governing fungal community assembly in Karst caves.

## 1. Introduction

Understanding the assembly processes and mechanisms that shape community diversity, biogeography, succession, and function is a prominent topic of study in ecology, particularly in microbial ecology. However, most of these processes are still not well-understood or are highly controversial [1,2,3,4]. Deterministic processes, often referred to as niche-based theory, propose that community structure is governed by deterministic factors including biotic interactions (e.g., competition, predation, mutualisms, and trade-offs), and abiotic factors such as environmental conditions (e.g., pH, temperature, salt, and moisture). These factors are believed to shape the species traits and interactions that ultimately determine community composition. On the other hand, the neutral theory posits that community structures are driven by stochastic processes including birth, death, colonization, extinction, and speciation [2,3,5]. According to this theory, community is primarily determined by stochastic processes, rather than by abiotic or biotic factors. It is widely accepted that both deterministic and stochastic factors play a simultaneous role in community assembly, whereas their relative importance remains a topic of ongoing debate [2,3,4].

Caves are unique ecosystems with distinct characters that differentiate them from outside environments, making them ideal conditions in which to test community assembly theories [6,7]. However, the relative contribution of stochastic and deterministic processes to the assembly of fungal communities in caves has not been thoroughly investigated. Previous studies on caves have mainly focused on fungal diversity, revealing a fabulous diversity with important ecological roles [6,8,9,10,11,12,13,14,15,16]. Additionally, some studies have identified various factors that influence fungal communities in cave ecosystems, including carbon resources, nutrient availability, geographic location, substrate type, and microclimate [11,16,17,18,19,20,21,22,23]. Furthermore, it is reported that fungal communities in caves can also be influenced by communities outside of caves [9,21,24,25]. However, despite several attempts to explore the controlling factors of fungal communities in caves, none of them have tried to dissert the relative contribution of stochastic and deterministic processes to the assembly of fungal communities therein.

By collecting a large number of air, rock, sediment, and water samples from eight representative Karst caves in Southwest China, Zhang et al. systematically investigated the diversity, composition, and controlling factors of fungal communities in Karst caves using a high throughput sequencing method [21]. In the current study, data from the study of Zhang et al. [21], which provided a comprehensive overview of fungal communities in caves on a large geographic scale, were used to resolve the assembly patterns of fungal communities in caves. The findings of Zhang et al. revealed that geographic location and substrate type were two major determinants of fungal community composition in caves [21]. Accordingly, we hypothesized that, on a single cave scale, fungal community assembly is driven by deterministic factors, while on a larger geographic scale, stochastic factors play a more significant role. Given the crucial role of substrate in shaping fungal communities in caves, as highlighted by Zhang et al. [21], the dataset was divided into four groups, air, rock, sediment, and water, in order to explore community assembly patterns and the relative influence of each variable on community structures on large scale.

## 2. Materials and Methods

### 2.1. Sample Collection, Processing and Data Processing

Samples and molecular data used in this analysis were acquired from our previous study [21]. In brief, eight representative caves in Southwest China were selected, these were accessible but relatively primitive, with few tourists. In each cave, air, rock, sediment, and water samples were collected. For each substrate, duplicate samples of four to six sampling sites with equal distances between adjacent sites in one cave were selected based on cave length, of which, one was for DNA extraction and the other one was a backup. Aerial fungi in two cubic meters of air samples at each site were filtered, by a mesh bracket-supported 90 mm diameter polyester fiber microfiltration membrane with 0.22-μm pore size, using a Millipore air sampler MAS-100Eco (Merck Millipore, Darmstadt, Germany) with a filter speed of 100 L per minute (L/min). Fungi in water at each site were filtered from two liters of water using the same membranes with a 60 mm diameter when water was available. For sediment samples, shallow sediments from three sites at each location were mixed thoroughly and thirty grams were kept. Five pieces of rock in different orientations at each location were collected. In total, 26 air samples, 36 rock samples, 36 sediment samples, and 19 water samples were collected. All samples collected were preserved and transferred at 4 °C as quickly as possible and stored at −80 °C in the laboratory until processing. Spatial variables, including longitude, latitude, altitude, and length of caves, and environmental variables, including temperature, humidity, and physiochemical properties, such as pH, moisture, total organic carbon (TOC), total nitrogen (TN), total phosphorus (P), calcium (Ca), copper (Cu), iron (Fe), potassium (K), magnesium (Mg), sodium (Na), and zinc (Zn), were measured as described by Zhang et al. [21].

Total genomic DNA was extracted using a FastDNA Spin Kit (MP Biomedicals, Solon, OH, USA) according to the manufacturer’s instructions. The ITS1 rDNA region of each sample was amplified using the primer pair ITS1F/ITS2 [26] with equal polling of three replicates. Amplicons were sequenced using the Illumina Miseq PE250 sequencing platform (Illumina Inc., San Diego, CA, USA) at the Majorbio Bio-Pharm Technology Company (Shanghai, China). Generated raw amplicon data were archived in the Sequences Read Archive (SRA) at the National Centre for Biotechnology Information (NCBI) with BioProject ID PRJNA486070.

Raw sequences were paired using FLASH (v1.2.11) [27], and then quality filtered using QIIME (v1.9.1) [28]. Subsequent chimera checking and removal were performed using Mothur (v1.43.0) [29]. OTUs were clustered at 97% similarity after sequence deduplication and sorting using USEARCH (v9.1.13) [30,31]. The taxonomic assignment of OTUs were performed by BLASTn (v2.6.0+) [32] searches on each representative sequence against a unified system for DNA-based fungal species linked to the classification (UNITE) and international nucleotide sequence databases (INSDC) fungal ITS databases (version released on 31 June 2017).

### 2.2. Statistical Analyses

#### Fungal Community Assembly Patterns 

To quantify the relative importance of stochastic and deterministic processes that drive fungal community assembly, beta nearest-taxon index (βNTI) and Bray–Curtis-based Raup–Crick (RC_Bray_) were calculated using Rscript “bNTI_Local_Machine.r” written by Stegen et al. [2] based on phylogenetic distance and OTU abundance. The βNTI is the number of standard deviations of the beta mean nearest taxon distance from the mean of the null distribution [2]. The RC_Bray_ value was used to further resolve the pairwise comparisons that were assigned to stochastic processes [2,3,26]. βNTI values between −2 and 2 indicate a dominance of stochastic processes, whereas those smaller than −2 or larger than 2 indicate that deterministic processes (i.e., homogeneous selection and heterogeneous selection) play a more important role in community assembly than stochastic processes [2,3,33]. When βNTI values are smaller than 2, RC_Bray_ < –0.95 and RC_Bray_ > 0.95 indicate a relative dominant influence of homogenizing dispersal and dispersal limitation, respectively, and RC_Bray_ values < 0.95 represent a crucial role for “Undominated” assembly, including weak selection, weak dispersal, diversification, and/or drift [3,33,34]. Furthermore, Pearson’s correlation coefficients and *p*-value were calculated to explore the associations between βNTI values and changes in environmental variables in different samples. Meanwhile, Pearson’s correlation coefficients between fungal community Bray–Curtis similarities and geographic distance between samples were calculated to determine the spatial predictors of fungal community composition. The geographic distance (in km) between each sample, i.e., the straight-line distance between the sampling points, was calculated using the R package geosphere (v1.5) [35] based on the longitude and latitude coordinates of each sampling site.

A variation partition analysis (VPA) using distance-based redundancy analysis (db-RDA) was performed to determine the relative proportions of community variations that can be explained by spatial (latitude, longitude, and sediment depth) and environmental (MAT, MAP, salinity, pH, gravel proportion, TC, TOC, TN, N/NH_4_^+^, N/NO_3_^−^, TP, and TS) variables together [36]. To assess the relative roles of each variable on fungal community structure, the Spearman’s correlation coefficient between the Bray–Curtis distances of fungal communities in each substrate, and the Euclidean distances of the spatial and environmental variables was calculated using the Mantel test in vegan package (v2.5) [36] based on 9999 permutations.

## 3. Results

In total, 33,601 OTUs with 22,635 rare OTUs (≤5 reads) were clustered at a 97% similarity from 6,618,705 clean reads. Among the non-rare OTUs, 10,769 OTUs, accounting for 97.4% of the clustered reads, were assigned into the fungal kingdom [21]. Before the following analyses, sequences in each sample were rarefied to 29,391, the lowest sequences number, and the samples outside of caves were removed.

### 3.1. Dispersal Limitation Plays a Crucial Role in Shaping the Overall Fungal Community in Karst Caves

Based on the OTU abundance and their phylogenetic distance, βNTI and RC_Bray_ indices were calculated to explore the relative importance of stochastic and deterministic processes in the assembly of the fungal community in caves (Figure 1). The results showed that, the average values (−1.15) and the majority (77.9%) of βNTI values in all samples were below 2 or above −2 (Figure 1a), suggesting that stochastic processes play a more significant role in community assembly compared to deterministic processes. Additionally, the majority (84.7%) of RC_Bray_ values in all samples were greater than 0.95 (Figure 1b), suggesting a crucial role of dispersal limitation in the assembly of cave fungal community. Consequently, according to the criteria described in the Methods section, the community assembly patterns in caves were categorized into five portions: homogeneous selection, heterogeneous selection, homogenizing dispersal, dispersal limitation, and “Undominated” processes. Among these patterns, dispersal limitation (66.0%) and homogeneous selection (20.9%) were the most crucial processes in controlling overall community assembly (Figure 1c). Similar trends were observed in rock, sediment, and water samples, where dispersal limitation (59.1–74.7%) and homogeneous selection (17.5–22.7%) were the dominant assembly processes for the fungal community. However, in air samples, the “Undominated” processes accounted for the majority (38.9%) of community assembly, making them the most significant factor. Dispersal limitation (31.8%) and homogeneous selection (23.6%) also played important roles in the assembly of the air community (Figure 1c).

The analysis of βNTI and RC_Bray_ values revealed varying assembly patterns in each cave (Figure 1d–f). In cave C1, the processes controlling community assembly were homogeneous selection (47.4%), followed by dispersal limitation (28.8%) and “Undominated” processes (19.9%). In cave C2, the most influential processes were dispersal limitation (43.1%), homogeneous selection (33.3%), and “Undominated” processes (17.2%). In Cave Y3, the dominant processes were “Undominated” processes (44.9%), followed by dispersal limitation (29.8%) and homogeneous selection (22.2%). In contrast, dispersal limitation played a crucial role in the community assembly in the remaining caves, particularly in caves G1, G3, and S7, where its contribution ranged from 68.7% to 73.7% (Figure 1f).

βNTI values in air samples did not show a significant correlation with any of the environmental variables. However, βNTI values in rock samples were significantly correlated with changes in temperature, TOC, Ca, K, and Mg. Additionally, βNTI values in sediment samples exhibited significant correlations witwith changes in temperature and TN (Figure 1g). Furthermore, Bray–Curtis similarity values of fungal communities in all, air, rock, sediment, and water samples exhibited significant negative correlations with the geographic distance, indicating a clear distance-decay relationship between them (Figure 2), and providing support for the significant effects of stochastic processes on the assembly of fungal communities in caves.

### 3.2. Environmental Selection Plays a Minor Role in the Fungal Community in Karst Caves

The VPA analysis demonstrated that only 7.0% of the overall fungal community could be explained by spatial variables, and environmental variables explained only 2.5% of the variations. Interestingly, both spatial and environmental variables together accounted for 0.6% of the variations. Consequently, the combined impact of spatial and environmental variables explained a total of 8.9% of the overall variation in the fungal community (Figure 3). Further analyses revealed that the explained variation in fungal community due to spatial and environmental variables was much higher in air samples (32.3%) compared to rock (9.6%), sediment (18.6%), and water samples (9.0%) (Figure 3).

In the case of air samples, the fungal communities were significantly influenced by latitude, longitude, altitude, and temperature, while air humidity and sample distance from the entrance did not show significant effects (Figure 4a). Similarly, fungal communities in water samples were significantly correlated with latitude, longitude, altitude, and temperature (Figure 4b). The rock communities, on other hand, were influenced by latitude, longitude, altitude, temperature, and air humidity (Figure 4c). For sediment communities, both spatial variables (latitude, longitude, and altitude) and environmental variables (temperature, air humidity, pH, TOC, TN, Ca, Cu, P, and Zn) had significant effects (Figure 4d). Interestingly, it was observed that latitude, longitude, altitude, and temperature had significant impacts on the fungal communities across all substrates. Specifically, latitude had the greatest effect on the air (Mantel’s r = 0.329, Mantel’s *p* = 0.0002) and water (Mantel’s r = 0.351, Mantel’s *p* = 0.0028) communities, while temperature and altitude played more important roles in the rock (Mantel’s r = 0.369, Mantel’s *p* = 0.0001) and sediment (Mantel’s r = 0.422, Mantel’s *p* = 0.0001) communities (Figure 4).

## 4. Discussion

Various fundamental biodiversity patterns have been observed in ecology [3,37]. However, the mechanisms and factors controlling these biodiversity patterns remain unclear or highly controversial [3,4]. In the current study, we disassembled the assembly patterns of the fungal community in caves based on the data reported in Zhang et al. [21]. The results suggested that dispersal limitation was the primary process controlling the assembly of the fungal community in caves on a large scale, while the predominant assembly processes in individual caves were different. Meanwhile, the influence of spatial and environmental variables on the community variations was found to be relatively low. The findings of this study expand our knowledge on the factors shaping the fungal community in caves.

### 4.1. Geographic Barriers between Caves May Lead to the Dispersal Limitation of the Overall Fungal Community in Karst Caves

In the current study, it was found that stochastic factors were the dominant drivers of fungal community assembly in Karst caves on a large scale. Stochastic processes have been recognized as controlling processes of microbial biodiversity in various studies [3,4,38,39,40]. The findings were further supported by the strong correlation observed between the similarity of fungal communities and geographic distance (Figure 2). The distance-decay correlation suggests that the community similarity tends to decrease with increasing geographic distance due to dispersal limitation and ecological drift [41]. Consistently, dispersal limitation played crucial role in community assembly on large scale (Figure 1c). Dispersal refers to the movement and successful establishment of individuals across space, either passively or actively [1,2,3]. Thus, if the movement or survival of organisms is restricted in a new location, they may exhibit dispersal limitation, resulting in more dissimilar community structures among communities [3]. While the microbial communities in Karst caves are influenced by communities outside of the caves, the caves themselves remain relatively closed ecosystems [21,42]. The caves investigated in this study were far away from each other, indicating great geographic barriers between caves. These great geographic barriers can contribute to geographic isolation, thereby leading to dispersal limitation of the fungal community in karst caves. The strong effects of dispersal limitation on the fungal communities in rock, sediment, and water samples may also be attributed to the strong geographic isolation.

It is noteworthy that the importance of “Undominated” processes in the community assembly of air samples is greater than that of dispersal limitation (Figure 1c). There are two possible explanations for the different predominant assembly processes observed between air samples and other substrates. First, compared to fungal communities in other substrates, the communities in air samples were more homogeneous along caves and were more influenced by the communities found outside the caves, due to airflow and animal activities [24,25,43]. According to the analyses of Zhang et al. [21], fungal communities in air samples were more similar than in other substrates, which can be attributed to the airflows found in most caves investigated. Several other studies on cave fungal communities in Europe found that bats might contribute to the increase in airborne fungal diversity and act as vectors for microscopic fungi [24,43]. In other words, the exchange of air communities between the cave and the outside environment was more frequent than in other substrates, resulting in more similar communities. Second, “Undominated” processes encompasses weak selection, weak dispersal, diversification, and drift [3,44]. However, the contributions of each pattern to “Undominated” processes were difficult to parse. Weak selection in “Undominated” processes may arise from influential selective forces that counteract each other or contrasting selection. For example, both homogeneous and heterogeneous selection on different taxa could lead to random phylogenetic patterns [3]. As potential support, the VPA analysis (Figure 3) and db-RDA analysis in Zhang et al. [21] revealed that environmental selection explained a higher degree of community variation in other substrates compared to air samples, except for water samples, in which the influence of environmental selection was the lowest. The low effect of environmental variables is possibly because the water in caves mostly comes from the above-ground environment [21].

The dominant processes controlling community assembly in single caves were observed to be in contrast to our hypothesis. Dispersal limitation was found to be predominant in six caves, while homogeneous selection and “Undominated” processes were dominant in the remaining two caves. This contrast may be attributed to the specific structure and conditions present in each cave. For example, some caves are strongly zonal, while others consist of several large halls. Additionally, the influences of external communities, as well as animal and human activities, on fungal communities differ among caves. For instance, Zhang et al. found that fungal communities in Karst caves were greatly influenced by fungal communities in external environments due to airflow exchange [21,45]. Meanwhile, bats and human activities might introduce foreign fungi and change the fungal community in caves [24,25,46,47,48]. However, assessing the effects of these factors on community assembly is challenging. On the other hand, it is important to exercise caution when interpreting dispersal limitation as evidence for stochastic processes, as both stochastic and deterministic factors can contribute to dispersal. Dispersal is considered stochastic when dispersal rates depend on the population size, meaning that more abundant species have a greater dispersal potential than less abundant species. Conversely, dispersal is deterministic when it is dependent on species traits and active status, or habitat features for establishment [1,3,5,49]. However, as it is difficult to identify dispersal traits, most studies, including ours, still treat dispersal as neutral [3,5]. Another interesting finding is the strong effect of “Undominated” processes, contributing 12.3% to as much as 44.9% to community assembly. This suggests a high proportion of assembly processes that cannot be decomposed in Karst caves [3,38].

### 4.2. Environmental and Spatial Factors Have a Relatively Weak Impact on the Fungal Communities in Karst Caves

Consistent with the findings of Zhang et al. [21], our study also showed that spatial and environmental variables had a stronger influence on community variations in air samples compared to other substrates (Figure 3), supporting our observation that environmental selection played a more prominent role in community assembly in air than in other substrates (Figure 1c). However, the explained variations in fungal community in caves were quite low in both our study and that of Zhang et al. [21]. This suggests that environmental and spatial factors have a relatively weak impact on shaping fungal communities. There are several potential explanations for this observation [5,21]. First, there may be some other crucial factors that were not considered in the current study. Second, methods such as VPA and db-RDA may not fully capture the relationships between microbes that significantly affect community structure [50,51]. Third, VPA might lead to incorrect prediction regarding the explained community variation, and therefore, the results should be interpreted in conjunction with other approaches, such as βNTI [3].

In conclusion, our study provides a preliminary quantification of the assembly patterns of fungal communities in Karst caves, based on our previous investigation of fungal communities in eight caves in Southwest China. We found that dispersal limitation plays a crucial role in shaping both the overall fungal community and specific communities in rock, sediment, and water samples. However, “Undominated” processes have greater influence on the air fungal community compared to dispersal limitation. Moreover, the dominant processes differ between each cave. VPA analysis revealed that the environmental selection plays a minor role in the fungal community in caves. Among the spatial and environmental variables examined, latitude, longitude, altitude, and temperature significantly affect the fungal communities across all substrates. Overall, our study provides a valuable insight into the mechanisms and factors that control the assembly of fungal communities in Karst caves, which may contribute to further exploration and conservation efforts for microbial communities in these unique environments.

## Figures and Tables

**Figure 1 jof-09-01013-f001:**
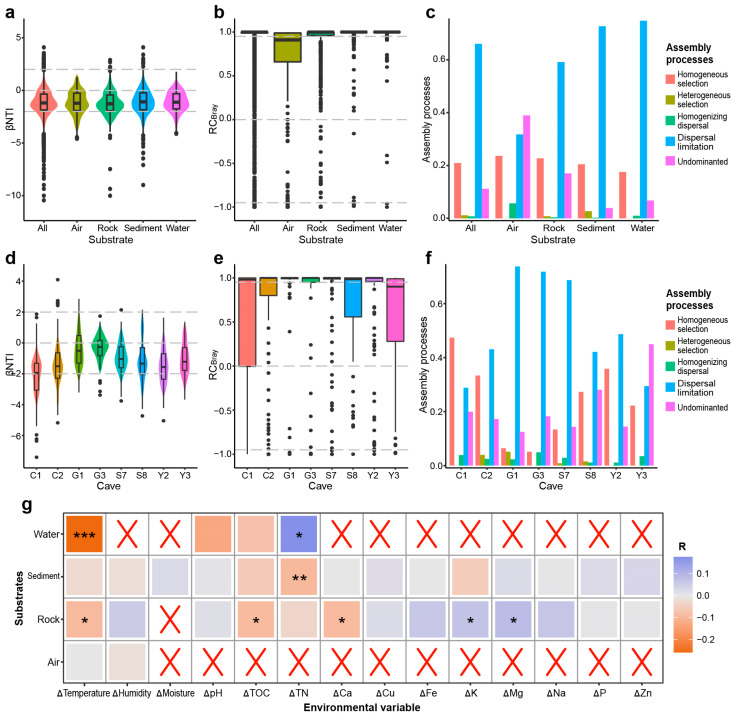
Assembly patterns of fungal communities in Karst caves. (**a**) β-nearest-taxon index (βNTI) of fungal communities in caves on a large scale. Horizontal dashed lines (βNTI values at 2 and –2): thresholds of significance. (**b**) Bray–Curtis-based Raup–Crick (RC_bray_) values of fungal communities in caves on a large scale. Horizontal dashed lines: RC_bray_ value at 0.95 and –0.95. (**c**) The percentage turnover of community assembly governed primarily by various deterministic processes, including homogenous and heterogeneous selection, and stochastic processes, including dispersal limitations and homogenizing dispersal, as well as “Undominated” processes. (**d**) βNTI of fungal communities in individual caves. (**e**) RC_bray_ values of fungal communities in caves on a large scale. (**f**) The percentage turnover of community assembly governed primarily by various deterministic processes. (**g**) The relationships between βNTI and changes in the environmental variables. Statistical significance is displayed as follows: * *p* ≤ 0.05; ** *p* ≤ 0.01; *** *p* ≤ 0.001.

**Figure 2 jof-09-01013-f002:**
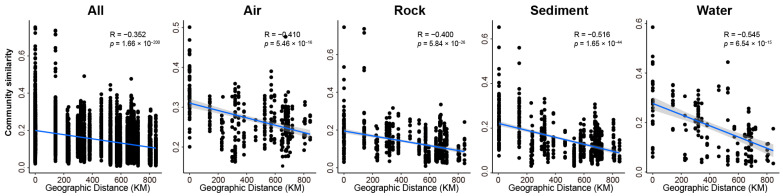
Distance-decay patterns of fungal community similarity and geographic distance. The shaded area around the lines covers the 95% confidence interval of the correlations. The associated correlation coefficients and *p*-values are shown in each panel.

**Figure 3 jof-09-01013-f003:**
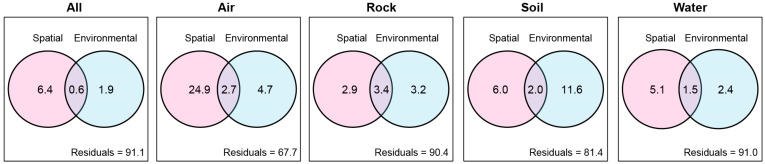
Effects of spatial and environmental selection together on fungal community. Variation partition analysis based on Bray–Curtis dissimilarity matrices, partitioning the relative contributions of spatial and environmental factors to fungal community structure.

**Figure 4 jof-09-01013-f004:**
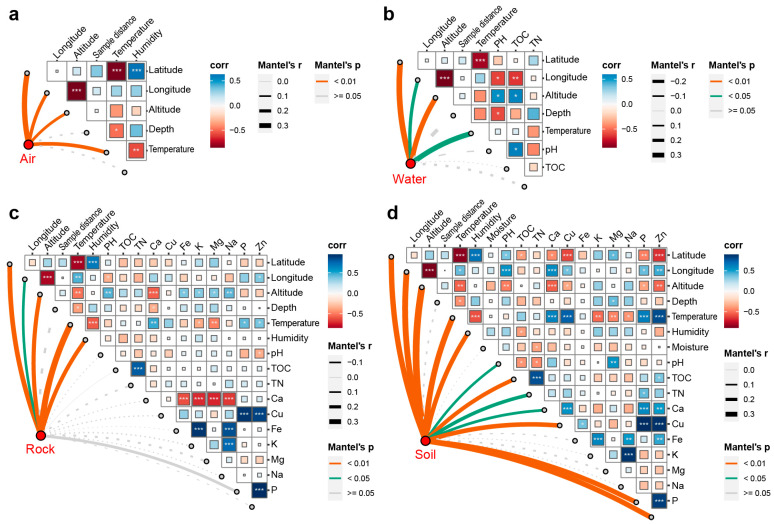
Effects of spatial and environmental variables on fungal community based on Mantel test. (**a**) Air samples, (**b**) water samples, (**c**) soil samples, (**d**) sediment samples. The red dots on the lower left corner of each panel represent communities in different substrates. The heatmaps on the top right corner show the relationships among spatial and environmental variables, in which, “*” represents correlation significance (* *p* ≤ 0.05; ** *p* ≤ 0.01; *** *p* ≤ 0.001). The lines linking the red dots and right heatmaps represent the effects of variables on fungal communities. The effect of each variable is displayed by the line width, and the significance is displayed by the line color.

## Data Availability

The raw data of metabarcoding was deposited to the Sequence Read Archive of NCBI: PRJNA486070.

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
