# Peer review of "Dispersal Limitation Controlling the Assembly of the Fungal Community in Karst Caves"

_jof, 2023, doi:10.3390/jof9101013_

Round 1

Reviewer 1 Report

It's a well-written and solid article. However, additional information on the caves, on the number of samples would have made the article more independent from the Zhang, Cai 2018 article (previous article by the authors).
The materials and methods  a lack information (also missing from article no. 18) is the diameter of the filter - the MAS 100Eco is normally designed for Petri dishes.
The way in which the replicates of the samples were processed

the Mantel test should be written with M and not m.

Author Response

1. It's a well-written and solid article. However, additional information on the caves, on the number of samples would have made the article more independent from the Zhang, Cai 2018 article (previous article by the authors).

Answer: Thanks for your positive comments and valuable suggestions on this article. More additional information on caves and samples have been added (Line 72–73 and 77–85).

2. The materials and methods a lack information (also missing from article no. 18) is the diameter of the filter - the MAS 100Eco is normally designed for Petri dishes.

The way in which the replicates of the samples were processed

Answer: Yes, MAS 100Eco is designed for Petri dishes. To filter air samples using this machine, we designed a special mesh bracket. The missing information was added in Line 76–78.

3. the Mantel test should be written with M and not m.

Answer: Corrected, thank you (Line 136).

Reviewer 2 Report

The manuscript (ID jof-2643047) submitted for review is written in correct scientific language and well prepared statistically and graphically. The detailed graphic design is impressive. Although I am very concerned about the following sentence in the methodology "... Samples and molecular data used in this analysis were acquired from our previous study [18]...". Why was this data not published in this manuscript? I ask the authors to respond to this issue.

Currently, several manuscripts have been written on the influence of microclimatic conditions on the number of fungi in underground ecosystems (e.g. 10.4311/2013MB0123 ; 10.1080/01490451.2014.907380 ; 10.5038/1827-806X.43.1.3 ; 10.1080/01490451.2017.1280860), microclimatic conditions and composition of rocks (e.g. 10.1007/s11629-016-4221-y) as well as bats (e.g. 10.1007/s00248-016-0763-3 ; https:// doi.org/10.3390/biology10070593). Please discuss these works in the Discussion section.

Overall, the manuscript looks good, but the above-mentioned issues must be clarified before it can be accepted for publication.

The language side of the work looks good - small corrections.

Author Response

1. The manuscript (ID jof-2643047) submitted for review is written in correct scientific language and well prepared statistically and graphically. The detailed graphic design is impressive. Although I am very concerned about the following sentence in the methodology "... Samples and molecular data used in this analysis were acquired from our previous study [18]...". Why was this data not published in this manuscript? I ask the authors to respond to this issue.

Answer: Thanks for your positive comments and valuable suggestions on this article. The raw data had been published in our previous study, while in that article, only the diversity, composition, and influencing factors of fungal community were analyzed. However, besides influences of environmental variables analyzed previously, stochastic processes were also important for fungal community. So, after that, I read a number of articles on stochastic and deterministic theories, and have a greater understanding in these ecology theories. In this study, we tried to analyze relative importance of stochastic and deterministic processes on fungal community.

2. Currently, several manuscripts have been written on the influence of microclimatic conditions on the number of fungi in underground ecosystems (e.g. 10.4311/2013MB0123 ; 10.1080/01490451.2014.907380 ; 10.5038/1827-806X.43.1.3 ; 10.1080/01490451.2017.1280860), microclimatic conditions and composition of rocks (e.g. 10.1007/s11629-016-4221-y) as well as bats (e.g. 10.1007/s00248-016-0763-3 ; https:// doi.org/10.3390/biology10070593). Please discuss these works in the Discussion section.

Answer: Discussion on listed articles and more related articles have been added (Line 49, 51, 272-276, 299-302), thank you.

3. Overall, the manuscript looks good, but the above-mentioned issues must be clarified before it can be accepted for publication.

Answer: Thanks for your suggestions, all issues mentioned have been clarified in Comment 1 and 2.

Round 2

Reviewer 2 Report

The current version of the manuscript has been significantly improved from the original version, and the authors have taken my comments into account. Good job!

Minor editing of English language required